# Statistical Modeling of the Early-Stage Impact of a New Traffic Policy in Milan, Italy

**DOI:** 10.3390/ijerph17031088

**Published:** 2020-02-08

**Authors:** Paolo Maranzano, Alessandro Fassò, Matteo Pelagatti, Manfred Mudelsee

**Affiliations:** 1Department of Statistics and Quantitative Methods (DISMEQ), University of Milano-Bicocca, 20126 Milano, Italy; 2Department of Management, Information and Production Engineering (DIGIP), University of Bergamo, 24044 Dalmine, Italy; alessandro.fasso@unibg.it; 3Department of Economics, Management and Statistics (DEMS), University of Milano-Bicocca, 20126 Milan, Italy; matteo.pelagatti@unimib.it; 4Climate Risk Analysis, 37581 Heckenbeck, Germany; mudelsee@climate-risk-analysis.com

**Keywords:** air pollution, oxides, traffic, state space, milan, area b, cross validation, policy intervention analysis, counter-factual, unobservable components

## Abstract

Most urban areas of the Po basin in the North of Italy are persistently affected by poor air quality and difficulty in disposing of airborne pollutants. In this context, the municipality of Milan started a multi-year progressive policy based on an extended limited traffic zone (Area B). Starting on 25 February 2019, the first phase partially restricted the circulation of some classes of highly polluting vehicles on the territory, in particular, Euro 0 petrol vehicles and Euro 0 to 3 diesel vehicles, excluding public transport. This is the early-stage of a long term policy that will restrict access to an increasing number of vehicles. The goal of this paper is to evaluate the early-stage impact of this policy on two specific vehicle-generated pollutants: total nitrogen oxides (NO${}_{x}$) and nitrogen dioxide (NO${}_{2}$), which are gathered by Lombardy Regional Agency for Environmental Protection (ARPA Lombardia). We use a statistical model for time series intervention analysis based on unobservable components. We use data from 2014 to 2018 for pre-policy model selection and the relatively short period up to September 2019 for early-stage policy assessment. We include weather conditions, socio-economic factors, and a counter-factual, given by the concentration of the same pollutant in other important neighbouring cities. Although the average concentrations reduced after the policy introduction, this paper argues that this could be due to other factors. Considering that the short time window may be not long enough for social adaptation to the new rules, our model does not provide statistical evidence of a positive policy effect for NO${}_{x}$ and NO${}_{2}$. Instead, in one of the most central monitoring stations, a significant negative impact is found.

## 1. Introduction

Air quality monitoring is one of the major challenges that European institutions jointly with national and local administrations are facing in terms of environmental protection. In particular, the 2008 European Air Quality Directive (AQD) 2008/50/EC [1] requires EU Member States to design appropriate air quality plans for zones where the air quality does not comply with the AQD limit values. In the last few decades, European countries implemented various modeling methods to assess the effects of local and regional emission abatement policy options on air quality and human health [2]. On the one side, they include scenario approaches, in which running a chemical-physical simulation model with and without a specific emission source allows for quantifying the impact on air quality levels [3,4]. On the other side, they also include more comprehensive and multidisciplinary approaches, such as Integrated Assessment Models (IAM), which combine simultaneously many features of the economy, society, and scientific findings. These models are based on the combination of multiple mathematical tools and allow for assessing the impact of environmental policies or to improve the air quality control system. Typical tools are the full cost-benefit analyses [5], in which abatement measures, costs, and benefits are expressed in monetary units, optimization, and spatial analysis [6,7].

In areas such as Northern Italy, where the industrial transition in the 1990s reduced coal burning and sulphur concentration, the large majority of environmental studies focus their attention on toxic pollutants that are produced by thermic vehicle engines and house heating plants. These are known to generate serious health effects [8]. Total nitrogen oxides (NO${}_{x}$), nitrogen dioxide (NO${}_{2}$), and particulates matters (PM${}_{10}$ and PM${}_{2.5}$) belong to this class.

According to the above EU rules, governments adopted standards and quantitative limits for pollutant emissions to make economic agents responsible and implement abatement policies. In particular, the maximum concentration for NO${}_{x}$ and NO${}_{2}$ is set to 40 μg/m${}^{3}$ annual average and 200 μg/m${}^{3}$ hourly not to be exceeded more than 18 times in a single year. Figure 1 represents the average concentration levels of NO${}_{2}$ in Europe for the year 2018. The Po basin in Northern Italy stands out as a heavily polluted area with difficulties in pollution management. The negative impact on society is not limited to health only. There is increasing evidence showing that bad air quality in general, and high NO${}_{2}$ concentrations in particular, impact the economy, including finance [9] and tourism [10].

The present paper analyses the introduction of the first phase of an air quality control policy in the municipality of Milan, which started on 25 February 2019 and directly acts on traffic rules. The administration defined an extended limited traffic zone, named Area B (https://www.comune.milano.it/aree-tematiche/mobilita/area-b), where the access and circulation for the most polluting vehicles, as well as those longer than 12 meters, have some partial restrictions, enforced by a monitoring system of entrance gates controlling each license plate and imposing a fine on unauthorized vehicles. The access prohibition concerns Euro 0 petrol vehicles and a large part of Euro 0, 1, 2, and 3 diesel vehicles, with specific exemptions for public transport, itinerant traders, and residents, and it is active from Monday to Friday during business hours (from 7:30 a.m. to 7:30 p.m.), except holidays. According to the municipality of Milan, the share of cars registered in the Milan metropolitan area and involved in the restrictions in 2019 is close to 17%, while the share of freight transport vehicles is around 53% [11]. Area B is a progressive policy divided into various phases, which will concern an increasing number of vehicle classes. In terms of NOx emissions, the administration expects a reduction of 4–5% per year until 2022 and a reduction of 11% between 2023 and 2026. The policy will be fully operative within October 2030.

Area B extends the previously existing limited traffic zone, Area C, which covers just the historical city centre. The physical coverage of the two restriction zones is represented in Figure 2, which highlights the arrangement of both within the city borders. Area B covers almost the entire area of the city, excluding extreme peripheral districts.

Statistical literature on air quality grew up sharply in the last decades. Two main statistical modeling directions have been developed. One has a focus on pollutants concentration and the other on human exposure. Regarding the latter, recent advances are based on crowdsourced data, such as smartphone data modeling [12]. Regarding pollutants concentrations, increasing attention is being given to latent component models; see, as an example [13] and for the problem of misalignment. In particular, the use of the INLA-SPDE approach for misalignment between pollutant concentration and epidemiological data [14] and PCA based methods with missing data [15].

When the territory under study is large and spatial correlation is important, spatio-temporal models are appropriate. See, for example, the multivariate state space approach of Calculli et al. [16], which is capable of handling jointly PM${}_{10},$ NO${}_{2}$ and weather variables, the approach of Menezes et al. [17] for modeling daily NO${}_{2}$ trends in Portugal. Moreover, the land-use regression model (LUR) under a state space approach has been used for modeling air pollution in Tehran [18]. Despite this growing spatial literature, time series analysis methods have been recently developed to understand the effect of meteorology on pollutant concentration [19], which will be the main focus of this paper.

The previous Milan limited traffic zone, known as Area C, has already been treated in literature by Fassò [20], who analyzed its introduction through spatio-temporal models, by Invernizzi et al. [21], who considered its impact on black carbon, and by Percoco [22] who considered its effect on traffic. Moreover, similar problems have been studied for London "sulphur-free zone" [23] and the "low emission zone" in Munich [24]. In Fassò [20], the author considered both particulates and nitrogen oxides and observed the presence of a more pronounced permanent reduction of the latter within the restricted area, despite the data showing a strong spatial variability depending on the type of pollutant. This is consistent with the known emissions pattern of particulate matters and nitrogen oxides. The latter are mainly primary gaseous pollutants and can be directly attributed to anthropogenic sources, such as car traffic and house heating. Moreover, from the so-called INEMAR emission inventory [25], in Milan province, 68% of NOx and only 41% of PM10 are due to road traffic. Hence, in this first study of Area B, we will take into account NO${}_{x}$ and NO${}_{2}$ and postpone the analysis of PM10 and PM2.5 to further research. To adjust for confounding factors, we will consider weather conditions in Milan, the main calendar events, and the concentration levels of oxides observed in neighbouring towns, as in a pseudo-treatment-control approach.

The study aims to identify and quantify variations in pollutant levels due to the above described Area B. Hence, the present paper will try to investigate and test the following two scientific hypotheses:
**Hypothesis** **1.**The introduction of Area B achieved significant changes in pollution concentration for the city of Milan;
**Hypothesis** **2.**The variation occurred homogeneously on the territory and the stations do not show spatial variability of the effects.

The first hypothesis aims to quantify the impact of the policy on pollution levels measured by several air quality stations scattered around the city and to assess whether this evidence is significantly supported by the data. The impact is evaluated both regarding the statistical significance of the estimates, the absolute magnitudes of the coefficients, and their signs. From the policy maker perspective, the expected coefficients should be negative, indicating a reduction effect on concentrations due to the car traffic restrictions. However, given the complexity of the phenomenon, a change of opposite sign cannot be ruled out either. The second research hypothesis is dedicated to the comparison of the estimates for the considered stations: the effect can be considered homogeneous when the sign and the magnitude of the coefficients for all the stations are similar.

The paper is structured as follows. Section 2 describes the dataset and the methodologies used for the analyses. In particular, we briefly explain the composition of both weather and air quality monitoring systems in Milan, available data sources, and metadata information. Then, we present the methodologies implemented for the preliminary analysis and the state space approach to time series analysis for air quality data. Section 3 reports and discusses the empirical results of the estimated models and their implications. Section 4 concludes the paper discussing the two research hypotheses in light of what emerged from the data analysis and gives some hints for future research developments.

## 2. Materials and Methods

In this section, we present the structure of the ARPA dataset and briefly introduce the methodologies implemented for the analyses. Section 2.1 introduces the data source for the Milan case study and the spatio-temporal structure of the data and provides a brief description of the variables taken into consideration. Section 2.2 designs the preliminary analysis, which introduces a temporal treatment-control experiment to highlighting the differences in concentration levels before and after the policy intervention. Section 2.3 gives a detailed overview of the use of state space models in time series analysis for the study of air quality data, including also a specific subsection for model selection and policy intervention.

### 2.1. Data

#### 2.1.1. Air Quality and Weather Monitoring Network in Milan

Data on pollution and weather conditions of Lombardy are collected from the Lombardy Regional Agency for Environmental Protection (ARPA Lombardia), which makes a large open data portal fully available to users (https://www.dati.lombardia.it/). The agency manages a diffuse monitoring system distributed among the regional territory and counting on hundreds of monitoring stations collecting intra-daily information on climate and pollution through sensors.

Installed within the borders of Milan are seven weather monitoring stations and five air quality monitoring stations. Air quality stations are classified according to a taxonomy system that identifies the type and function in the network. The stations Liguria (ARPA code 539), Marche (ARPA code 501), Senato (ARPA code 548), and Verziere (ARPA code 528) are urban traffic control units: sensors installed near important roads and intersections in order to accurately quantify the pollution generated by traffic. The station Città Studi (ARPA code 705) is instead of type urban background, that is, the station is located in such a position that the level of pollution is not mainly influenced by specific sources but by the integrated contribution of all the upwind sources at the station with respect to the predominant directions of the winds on the site [8]. The seven weather stations are Marche (ARPA code 501), Lambrate (ARPA code 100), Zavattari (ARPA code 503), Brera (ARPA code 620), Feltre (ARPA code 869), Rosellini (ARPA code 1327), and Juvara (ARPA code 502).

Figure 2 georeferences on the map the exact position of each station and allows for identifying the position with respect to Area B and Area C. Air quality stations are represented as blue points, while weather stations are the red points. Marche station (ARPA code 501), in the upper side of the map, is the only one to collect both weather and pollution data and is represented with a double label, the first one blue and the second red.

The spatial distribution of the stations is not uniform: air quality stations cover northern, eastern, central, and southern parts of the city, leaving the western districts uncovered; climate stations cover in detail the city centre and all the northern neighbours but are not installed in the south.

#### 2.1.2. Temporal Coverage, Pollutants, and Weather Measures

The analysis presented in this paper takes into account daily measures from 1 January 2014 to 30 September 2019, generating an overall sample of 2099 daily observations.

The whole, the monitoring system provides information about many urban pollutants, such as carbon dioxide, particulates, and oxides. All the pollutants are measured as μg/m${}^{3}$. As already stated in the Introduction, we focus our attention on concentrations of total nitrogen oxides (NO${}_{x}$) and nitrogen dioxide (NO${}_{2}$), which are mainly primary gaseous pollutants, hence considered as proxies of pollution emissions due to human activities, first of all car traffic.

Weather stations provide measures of local temperature (${}^{\circ }$C), rainfall (cumulated mm), humidity (%), global radiation (W/m${}^{2}$), wind speed (m/s), and wind direction. The wind direction is expressed in clockwise degrees from $0^{\circ }$ to $360^{\circ }$; for example, $90^{\circ }$ identifies winds going from east to west. To make results easier to interpret, we decide to aggregate the measurements on wind direction and speed by constructing a set of new variables that describe the average speed in the four quadrants of the compass rose. The Northeast quadrant (Q${}_{\mathrm{NE}}$) corresponds to degrees between 0 and 90, the Southeast quadrant (Q${}_{\mathrm{SE}}$) to degrees from 90 to 180, the Southwest quadrant (Q${}_{\mathrm{SW}}$) to degrees from 180 to 270 and the Northwest quadrant (Q${}_{\mathrm{NW}}$) to the remaining values lying between 270 and 360 degrees.

These measures will be used in the modeling part to capture local weather conditions specific to the city of Milan. Instead of using the data referring to the weather station closest to each air quality station, we preferred to aggregate each of the climate variables through a daily city average valid for each pollution station. In this way, the subsequent models will be fully comparable guaranteeing the maximum possible spatial coverage.

#### 2.1.3. Anthropogenic Activities

Human activities, and therefore the quality of the air we breathe, are often affected by calendar events that are recorded based on national, local, and religious holidays and weekends. Calendar effects are captured by a set of covariates, which identify the weekends and the main Italian holidays, both religious and secular. Holidays are collected in a dummy variable named *Holidays*, while the weekends are contained in a dummy variable named *WeekEnd*. Specific effects related to the behavior of people can be observed when holidays coincide with the weekend; therefore, we considered two terms of interaction between the two dummies. The interaction terms include those holidays that fall on Saturday, denoted as *Saturday:Holiday*, and those on which they fall on Sunday, which is *Sunday:Holiday*.

For a correct assessment of the effects of the traffic policy on pollutants concentrations in Milan, it is necessary to purify the estimates from any external weather or socio-economic effects overlapping with the policy and which may hence alter policy effects. This operation is accomplished by introducing a counter-factual term into the models represented by the pollution levels observed in other cities surrounding Milan. We considered seven important urban centres located in the Lombardy Po Valley area, which show socio-demographic and economic characteristics and weather conditions similar to Milan, but which cannot be directly affected by the limited traffic zone. These urban centres are Bergamo (East), Brescia (far East), Cremona (far Southeast), Lodi (Southeast), Pavia (South), Saronno (North) and Treviglio (East). As reported in Figure 3, the considered candidates cover a large territory surrounding Milan in all the directions while maintaining a sufficient distance to be considered independent in terms of traffic.

### 2.2. Methods: Average and Median Difference before and after the Policy

Figure 4 shows the temporal evolution of yearly average and median concentrations in the period preceding and following the entry into force of the policy for each control units located in Milan. According to the figure, starting from 2015, the city of Milan recorded a generalized reduction of concentration levels especially in peripheral areas, such as Marche and Liguria. Observed mean values for 2019 present a further reduction of concentrations rather apparently anomalous and significant. The comparison between the levels of NO${}_{x}$ and NO${}_{2}$ pairs for each station shows obvious common trends between the two pollutants both considering the annual average and median values. Averages and medians follow similar temporal patterns, but focusing on nitrogen oxides sensors, it is possible to note that the medians are significantly smaller than the averages, highlighting the heavy-tailed characteristic of the distribution (positive asymmetry) and the presence of extreme values. Following these facts, an interesting question to investigate is if, and how much, the greater difference observed in 2019 can be attributed to traffic restrictions, or if it is due to a general de-carbonization trend that the city is experiencing, or to weather variations not considered yet.

Before investigating the factors and causes that may have generated these sharp reductions, we perform a preliminary analysis of the concentration levels pre-and-post policy, in order to quantify the changes observed in 2019 both in Milan and in the other centres. Since air quality data present outliers and heavy-tail distributions given by extreme events, the only use of average values for central tendency estimation can provide misleading results. Therefore, we compare the central values obtained both considering the sample mean and the sample median, which is notoriously a more robust indicator if outliers occur [26,27].

The comparison is performed through the computation of two statistics based on the difference of central tendency indicators. The first statistic computes the difference between the average of the observations gathered after the policy intervention and the average of observations referring to the sub-period 2014–2018. The second statistics consists of computing the difference between median concentration levels observed in 2019 and before that year. The difference in average concentrations is denoted by $d_{AVG}$, whereas the difference in median concentrations is denoted as $d_{MED}$. Since both sub-periods are treated as independent of each other, from the statistical perspective, the statistics are assimilable to unpaired samples statistics.

Both statistics use the observations collected between 25 February and 30 September of each year, with a total length of 214 days. Approaches of this type can be framed in a context of treatment-control analysis, in which the data referred to the year 2019 constitute the treatment group, while the observations collected between 2014 and 2018 compose the control group. Control data refer to a 5-year-period; therefore, the concentrations measured on the same calendar day are aggregated into a single representative value calculated as the daily average concentration of the period 2014 to 2018. Denoting as $c_{ij}$ the observed pollutant concentration during the day *j*, where j=25February,…,30September, of the year *i*, where i=2014,…,2018, the average for a generic calendar day *j* is computed as $c_{j}=\frac{\sum_{i=2014}^{2018}c_{ij}}{5}$.

Let $U=\{u_{j},\;j=1,2,\ldots ,214\}$ be the treatment observations and $V=\{v_{j},\;j=1,2,\ldots ,214\}$ the control observations, the difference of averages is defined as $d_{AVG}=AVG\left(U\right)-AVG\left(V\right)$ and the difference of medians is calculated as $d_{MED}=MED\left(U\right)-MED\left(V\right)$, where AVG(.) is the temporal sample mean and MED(.) is the temporal sample median.

### 2.3. Methods: Time Series Modeling Using a State Space Approach

In this section, we discuss the time series models used to identify the policy effect, the estimation algorithms, and the related inference. Firstly, we introduce a brief description of the basic structural model (BSM) using a state space approach for time series analysis and the estimation algorithm based on the Kalman filter [28,29]. Then, we present a three-step procedure used to select the most representative model in terms of predictive power and quality of fit. As a last step, we explain how the policy intervention is included in the models and how it should be interpreted.

#### 2.3.1. Basic Structural Model for Air Quality Data

According to their physical characteristics, air pollution concentrations time series are often characterized by seasonality, high persistence [30,31], strong right skewness with uni-modal distribution, and scale invariance [32]. Therefore, we analyze the concentrations using the basic structural model [33,34] augmented by deterministic regressors for weather conditions and socio-economic features.

BSM is defined as a simple unobservable components model composed by local linear trend (LLT), stochastic seasonality, and irregular (white noise) component. LLT describes both the temporal evolution of the series level and its slope, while the seasonal component aims to capture cyclical behaviors given by natural and anthropogenic phenomena. We modeled the seasonal component using a trigonometric form for daily data, hence with period s=365, and considering only a few harmonics given the very regular and almost deterministic behavior of the series. This fact avoids the risk of a model over-parametrization.

Let $\left\{y_{1},y_{2},\ldots ,y_{n}\right\}$ be the time series of the observed pollution concentrations in logarithmic scale, the state space form of BSM without regressors is composed by the following equations:(1)$$y_{t}=\mu_{t}+\gamma_{t}+\varepsilon_{t}\;,$$
where $\varepsilon_{t}∼N\left(0,\sigma_{\varepsilon }^{2}\right)$ is the measurement error and
(2)$$LLT\;\left(Level\right):\;\mu_{t}=\mu_{t-1}+\beta_{t-1}+\eta_{t}\;,\;\;\eta_{t}∼WN\left(0,\sigma_{\eta }\right)\;,$$
(3)$$LLT\;\left(Slope\right):\;\beta_{t}=\beta_{t-1}+\zeta_{t}\;,\;\;\zeta_{t}∼WN\left(0,\sigma_{\zeta }\right)\;,$$
(4)$$Stochastic\;seasonality:\;\gamma_{t}=\sum_{j=1}^{k}\gamma_{j,t}\;,$$
where $k\leq \left\lfloor \frac{s}{2}\right\rfloor$ is the number of included harmonics and $\gamma_{j,t}$ is the non-stationary stochastic cycle
(5)$$\left[\begin{array}{c} \gamma_{j,t}\\ \gamma_{j,t}^{*} \end{array}\right]=\left[\begin{array}{cc} cos(2\pi j/s) & sin(2\pi j/s)\\ -sin(2\pi j/s) & cos(2\pi j/s) \end{array}\right]\left[\begin{array}{c} \gamma_{j,t-1}\\ \gamma_{j,t-1}^{*} \end{array}\right]+\left[\begin{array}{c} \omega_{j,t}\\ \omega_{j,t}^{*} \end{array}\right]\;,$$

$\omega_{t}∼WN\left(0,\sigma_{\omega }^{2}\right)$ and $\omega_{t}^{*}∼WN\left(0,\sigma_{\omega }^{2}\right)$ are white-noise processes with mean zero and variance $\sigma_{\omega }^{2}$.

Equation (Equation 1) is called measurement equation and describes the evolution of the observed series as the sum of the underlying components, while Equations (Equation 2)–(Equation 4) are named transition equations. Equations (Equation 2) and (Equation 3) compose the LLT and describe respectively the unobservable processes of the level and the slope, whereas Equation (Equation 4) describes the trigonometric seasonality evolution. Weather, socio-economic factors, and policy intervention will be included in the models adding a set of deterministic components to the measurement Equation (Equation 1). Since the BSM with Gaussian errors belongs to the class of Gaussian linear models, the estimation step has been performed using the Kalman Filter algorithm, an iterative procedure, which allows estimating simultaneously the unobservable components and the model’s parameters by maximizing the Gaussian likelihood function.

When dealing with Gaussian linear state space models, the parameters estimated using a maximum likelihood (ML) approach inherit the asymptotic properties of ML estimators [29]. The distribution of the MLE is asymptotically approximated using a Gaussian distribution, which allows deriving the usual asymptotic confidence intervals and t-tests for significance. Assuming a significance level of 5%, the estimates are statistically significant if the standardized value lies outside of the interval [−1.96,1.96], obtained using the quantiles of a Standard Normal distribution. Moreover, since the dependent variable is expressed in logarithmic scale, the coefficients have to be interpreted as relative increases or decreases in concentration levels due to a unitary increase in the explanatory variable.

#### 2.3.2. Three-Step Model Selection

We now propose a three-step procedure for model selection, which considers multiple rules based on cross-validation, information criteria, and stepwise regression. To avoid estimation bias due to the policy introduction, all the steps are computed using only the observations before the introduction of Area B that is, from 1 January 2014 to 24 February 2019.

Step 1 is designed for selecting the most predictive seasonal component, defined in Equation (Equation 4), comparing different model specifications, which consider a varying number of harmonics *k* for the trigonometric function. Specifically, we fit 10 alternative models for each station: in each of them, the trigonometric seasonality is modeled by an increasing number of harmonics ranging from k=1 to k=10. The use of an increasing number of sinusoids, in our case up to 10, allows the modeling of complex seasonality with strong variations within short periods, but at the same time increases the model complexity.

Once the seasonal component has been selected, step 2 introduces in Equation (Equation 1) a counter-factual term $x_{t}$ able to capture weather and socio-economic factors common to the Po basin and affecting the air quality of Milan. In our approach, the counter-factual candidates are the time series introduced in Section 2.1.3 and which refer to the measurements of pollutant concentrations in seven important cities around Milan. The new measurement Equation can be written as follows:(6)$$y_{t}=\mu_{t}+\gamma_{t}+\theta x_{t}+\varepsilon_{t}\;,$$
where $x_{t}$ is the logarithm of the counter-factual time series and θ is its coefficient, $\mu_{t}$ follows Equations (Equation 2) and (Equation 3), and $\gamma_{t}$ follows the specification obtained by step 1. The expected sign of θ is positive: higher levels of air pollution should correspond to high values in nearby cities due to similar conditions.

In step 3, we identify the best subset of calendar events and weather covariates, capturing residual variations not yet covered by the counter-factual or by the latent components. These residual variations are estimated by the smoothed observation disturbances from Equation (Equation 6) that is ${\hat{\varepsilon }}_{t}$, and describe residual patterns that have not been explained by the persistence of series, the seasonality or characteristics common to nearby territories of the region.

Relevant weather and calendar covariates are selected through a backward-forward stepwise regression algorithm, which uses as a starting model the auxiliary linear regression expressed in Equation (Equation 7). The equation represents the full model which sets up the smoothed observation errors ${\hat{\varepsilon }}_{t}$ as dependent variable and the weather conditions and calendar events as covariates:(7)$$\begin{array}{cc} {\hat{\varepsilon }}_{t} & =\tau_{1}Holidays+\tau_{2}WeekEnd+\tau_{3}Saturday:Holidays+\tau_{4}Sunday:Holidays\\ \; & \;+\tau_{5}Temperature+\tau_{6}Rainfall+\tau_{7}Radiation+\tau_{8}Humidity\\ \; & \;+\tau_{9}WindSpeedQ_{\mathrm{NE}}+\tau_{10}WindSpeedQ_{\mathrm{NW}}+\tau_{11}WindSpeedQ_{\mathrm{SW}}+\tau_{12}WindSpeedQ_{\mathrm{SE}}\\ \; & \;+e_{t}\;, \end{array}$$

The stepwise regression is set up twice for each station: in one case, it selects the model according to the Akaike’s Information Criterion (AIC), while, in the other, it uses the Bayesian Information Criterion (BIC). The algorithm starts estimating the full model and computes the AIC or the BIC. Iteratively, it drops out the predictors one at a time; at each step, it computes the new information criterion and considers whether the criterion is improved by adding back in a variable removed at a previous step. The procedure ends when the reintroduction of each omitted variable does not improve the information criteria.

In the first two steps, we select the seasonal component and the counter-factual term by fitting and comparing alternative models based on Equations from (Equation 1) to (Equation 4) according to their predictive power and their ability to adapt adequately to the observed data. The first principle, which tests the predictive power of the models, relies on the minimization of the cross-validated mean square forecasting error (MSFE) evaluated for up to 10-step-ahead forecast horizon, that is, ${\hat{y}}_{t+h}\;\forall \;h=1,2,\ldots ,10$, while the second compares the models in terms of estimation quality. The latter computes both corrected Akaike’s Information Criteria (AICc) and BIC intending to select the model that minimizes both. To identify a unique model for all the stations located in Milan, we proceed to a global comparison, both graphical and analytical, of the two blocks of indicators, giving attention to the overall performances and not focusing only on individual outputs.

According to the cross-validation principle for time series [35,36], we split the full time series into two subsets: a training set for model estimation and a test set for evaluating the out-of-sample forecast performances. The training set includes all the measurements until 24 February 2018, while the test set contains observations relative to the sub-period 25 February 2018–24 February 2019, for a total count of 365 out-of-sample observations. The exclusion of observations after the start of the traffic restrictions makes it possible to obtain unbiased estimates of the policy effects avoiding overlapping with other unidentified factors. Before starting the iterative loop, the model to evaluate is estimated just on time using the observations included in the original training set. At the end of the estimation, the cross-validation algorithm is iteratively implemented as follows. For each iteration, the algorithm extracts the first ten observations available in the test set, generating a forecasting set, and computes three quantities: the 1-to-10 step-ahead forecasts that is ${\hat{y}}_{t+h}\;\forall \;h=1,\ldots ,10$, the forecast errors ${\hat{y}}_{t+h}-y_{t+h}$ and the quadratic forecast errors ${\left({\hat{y}}_{t+h}-y_{t+h}\right)}^{2}$. The first out-of-sample observation is discarded and the set of forecasting observations is updated right-shifting the forecast horizon by 1 unit and adding the new observation. These operations are repeated for a number of times equal to the length of the test set, in our cases 365 times. The algorithm returns the output of 365 different sequences of 1–10 step-ahead forecasts; for each step-ahead h=1,2,…,10, the MSFE is calculated as $MSFE\left(h\right)=\frac{\sum_{j=1}^{365}{\left({\hat{y}}_{t+h}-y_{t+h}\right)}^{2}}{365}$.

#### 2.3.3. Policy Intervention Analysis

The introduction of new rules or limitations to individual behaviors can lead to the co-existence of multiple effects with different structure, such as simultaneous immediate changes and adaptive changes that take a long time before visible effects occur. Take into consideration that this fact leads to implement intervention analysis, which includes both permanent and transitory effects. Further details and examples of ARMA-like transfer function applied to intervention analysis are available in Pelagatti ([29]).

The policy intervention is modeled through the combination of two individual effects: (1) a permanent effect, estimated by $\delta_{1}$ that measures the level shift of pollutant concentrations given by the treatment and modeled as a step dummy, which is $D_{1t}$, which assumes a value equal to 1 starting from 25 February 2019; (2) a transitory effect, estimated by $\delta_{0}$ and evolving according to a first-order difference dynamics of the type
(8)$$w_{t}=\lambda w_{t-1}+\delta_{0}D_{2,t},\;$$
where $D_{2,t}$ is a impulse dummy, which assumes value equal 1 for 25 February 2019 and 0 otherwise and λ measures the persistence of the transitory effect. The sum of the two effects returns the total effect, which expresses the estimated overall reduction or increase in air pollutant levels generated by the policy. The measurement equation after the three-step model selection and augmented by the policy intervention is eventually expressed as follows:(9)$$y_{t}=\mu_{t}+\gamma_{t}+\theta x_{t}+Z_{t}\Phi +\delta_{1}D_{1t}+w_{t}+\varepsilon_{t}\;,$$
where $y_{t}$ is the logarithm of pollution concentrations in one of the stations in Milan, $x_{t}$ is the logarithm of pollution concentrations in the optimal counter-factual station, $\mu_{t}$ is the LLT evolving according to Equations (Equation 2) and (Equation 3), $\gamma_{t}$ is the optimal seasonal component selected in step one, $Z_{t}$ is a matrix containing the set of optimal subset of weather and calendar covariates selected in step 3, and Φ is the associate vector of coefficients.

#### 2.3.4. Software

All the statistical computations and figures have been carried out using the statistical software R [37]. For state space models estimation, the *KFAS* package [38] was used. Cross-validation, forecasting, and model selection codes have been developed by the authors. The graphic elaborations were obtained by using the packages *ggplot2* [39] and *sf* [40].

## 3. Results

In this section, we present and comment on the empirical results relating both to the differences between pre-and-post policy averages and medians and to the policy intervention analysis for the Milan Area B case study. Section 3.1 shows the variations in concentration levels of NO${}_{x}$ and NO${}_{2}$ for all the stations installed in Milan and for the other seven cities around it. Section 3.2 presents the model selection results, the values of the selection criteria, and the final model specifications. Section 3.3 reports the empirical estimates of the policy effects obtained through the basic structural model augmented by the policy intervention.

### 3.1. Average and Median Differences

Empirical differences of concentrations levels for all the considered stations are presented in Table 1. For both nitrogen oxides and nitrogen dioxide, using the difference of mean and median respectively, it reports the estimates of the difference between the average (the median) concentrations for the year 2019 and the average (the median) concentrations of the same period for the years 2014–2018.

The estimates highlight large negative differences in oxides concentrations between 2019 and the period 2014–2018, both in the metropolitan area of Milan and in almost all the surrounding towns. Particularly heavy reductions, and similar to those in Milan, were recorded in the cities of Bergamo (East) and Pavia (South). The simultaneous abatement inside and outside Milan confirms the presence of a general decreasing trend in the aggregate levels of pollutants for the Lombardy Po basin as already indicated by the previous figures.

The differences registered in Milan are relevant both in suburban districts, such as the stations Liguria (West) and Marche (North), and in the historical centre at the Senato station. For those monitoring stations, the reductions are larger than 16 μg/m${}^{3}$ for NO${}_{x}$ and 10 μg/m${}^{3}$ for NO${}_{2}$. In general, the differences between the averages and between the medians are quite similar, but in many stations, the reductions for the medians are stronger than the average differences. This fact is related to the skewed and non-symmetric characteristics of the distribution involved, as shown also in Figure 4. The above considerations on average and median pollution abatement are valid for both pollutants, in fact, the stations where the greatest differences are recorded for nitrogen oxides are the same for nitrogen dioxide.

These preliminary results do not allow for identifying the causes of the reductions and to state if they depend on common causes related to the environment and climatic factors or if they have been generated by the introduction of the new policy in Milan. The next section will attempt to investigate the variations through the modeling of possible environmental and anthropogenic factors able to influence the air quality of the city.

### 3.2. Model Selection

#### 3.2.1. Step 1: Detection of the Seasonal Components

Results relative to the first step of model selection are summarized in Figure 5 and Figure 6, which show the evaluation criteria for all the stations. For each station, the plots are organized in paired-panels; the left panel represents the 10-steps-ahead MSFE as a function of the forecast horizon and the number of harmonics modeling the seasonality (scale colour); the right panel shows the AICc–BIC pairs for each model. The optimal number of harmonics to model the seasonality is identified as the one that evaluates the minimum pair of AICc and BIC and returns the lowest MSFE curve. Both the estimates for NO${}_{x}$ and NO${}_{2}\;$ for the city of Milan agree unanimously in the selection of the model in which the seasonality is composed by a single harmonic (k=1); therefore, it can be rewritten as
(10)$$Optimal\;seasonal:\;\gamma_{t}=\gamma_{1,t}\;,$$
where $\gamma_{1,t}$ is
(11)$$\left[\begin{array}{c} \gamma_{1,t}\\ \gamma_{1,t}^{*} \end{array}\right]=\left[\begin{array}{cc} cos(2\pi /365) & sin(2\pi /365)\\ -sin(2\pi /365) & cos(2\pi /365) \end{array}\right]\left[\begin{array}{c} \gamma_{1,t}\\ \gamma_{1,t}^{*} \end{array}\right]+\left[\begin{array}{c} \omega_{1,t}\\ \omega_{1,t}^{*} \end{array}\right]\;,$$

$\omega_{t}∼WN\left(0,\sigma_{\omega }^{2}\right)$ and $\omega_{t}^{*}∼WN\left(0,\sigma_{\omega }^{2}\right)$.

#### 3.2.2. Step 2: Detection of the Counter-Factual Component

After selecting the seasonal component, we proceed to the selection of the counter-factual term. Estimates are summarized in Figure 7 and Figure 8, which show the results for NO${}_{x}$ and NO${}_{2}$. The plots are graphically organized like those related to step one, with the difference that the MSFEs and the AICc-BIC pairs are functions of one of the seven counter-factual candidates instead of the number of harmonics. The selection criteria follow the same rules used for the previous step.

The search for the optimal counter-factual term requires greater attention and detail than in the previous step as the minimizers are not unique. According to the plots, there is a restricted set of stations that are good candidates for the counter-factual role. The set includes the following cities: Treviglio, Pavia, Saronno, and Cremona. In particular, Pavia’s station achieves one of the best forecast and fitting performances for almost all the stations in Area B for both NO${}_{2}$ and NO${}_{x}$. Based on this last consideration, we select as the counter-factual term for future models the air quality monitoring station of Pavia, located South to Milan. Therefore, the final specification of the basic structural model will include as counter-factual term the logarithm of the concentrations in Pavia, $x_{t}=log\left(Pavia_{t}\right)$.

#### 3.2.3. Step 3: Detection of the Weather and Calendar Factors

The last step of model selection aims to select the optimal subset of local weather and calendar regressors after having selected the optimal unobservable components, common weather, and socio-economic factors, captured by the counter-factual. For each station and pollutant, the best models are reported in Table 2 and Table 3.

As expected, BIC-based models, being more parsimonious, retain fewer covariates than AIC-based models. Following this fact, we will use the BIC-selected models, but we now discuss some details about the selection process. Concerning the calendar, both criteria include in almost all cases the holidays, week-end, and Sunday holidays effects. AIC suggests adding also the interaction term between Sunday and holidays. Even if the interaction between Saturday and holidays is not always included, the final model will take into account the full set of calendar events and their interactions.

Regarding weather covariates, except for the wind speed, none of the others is included within the final models. Winds blowing from Southwest (Q${}_{\mathrm{SW}}$) are always selected and those coming from Northwest (Q${}_{\mathrm{NW}}$) are often included, and hence kept in the final model. Moreover, temperature, rainfall, solar radiation, and humidity are considered only by the AIC. This fact can be explained by the presence of the counter-factual, which captures not only common characteristics in terms of human behaviour and air quality conditions but also homogeneous climatic conditions common to all the areas considered.

#### 3.2.4. Final Model Specification

Based on the results of the three-step model selection procedure, the final specification of the BSM augmented by the policy intervention can be expressed using the following model:(12)$$\begin{array}{cc} Measurement:\;y_{t} & =\mu_{t}+\gamma_{t}+\theta x_{t}+\phi_{1}Holidays+\phi_{2}WeekEnd\\ \; & \;+\phi_{3}Saturday:Holidays+\phi_{4}Sunday:Holidays\\ \; & \;+\phi_{5}WindSpeedQ_{\mathrm{SW}}+\phi_{6}WindSpeedQ_{\mathrm{NW}}\\ \; & \;+\delta_{1}D_{1t}+w_{t}+\varepsilon_{t}\;, \end{array}$$
where $\varepsilon_{t}∼N\left(0,\sigma_{\epsilon }^{2}\right)$
(13)$$Seasonal\;component:\;\gamma_{t}=\gamma_{1,t}\;,$$
(14)$$Level:\;\mu_{t}=\mu_{t-1}+\beta_{t-1}+\eta_{t}\;\;\eta_{t}∼WN\left(0,\sigma_{\eta }^{2}\right)\;,$$
(15)$$Slope:\;\beta_{t}=\beta_{t-1}+\zeta_{t}\;\;\zeta_{t}∼WN\left(0,\sigma_{\zeta }^{2}\right)\;,$$
(16)$$Transitory\;policy:\;w_{t}=\lambda w_{t-1}+\delta_{0}D_{2,t}\;,$$
where $y_{t}$ is the logarithm of pollution concentrations in one of the stations located in Milan and $x_{t}=log\left(Pavia_{t}\right)$ is the logarithm of pollution concentrations in Pavia.

### 3.3. Basic Structural Model and Policy Intervention

In this section, we show the numerical results obtained using state space modeling to estimate both permanent and transitory effects generated by the introduction of Area B controlling for local weather conditions, anthropogenic effects, and common areal trends.

The maximum likelihood estimates of the coefficients and the components’ variances for the five air quality monitoring stations installed in Milan are reported in Table 4 and Table 5. The results appear to be coherent both for nitrogen oxides and nitrogen dioxide. First, the models identify statistically significant and positive coefficients for the counter-factual term, highlighting its capability to capture socio-economic and climatic factors common to neighbouring areas and coherent with the expected sign. Second, weekends and holidays exert a negative effect on concentration levels probably due to the reduction in the movements and productive activities of the city in those days. Their interactions are almost everywhere not statistically significant but with a positive sign and always less than the sum of the individual effects of the weekend and holidays. This fact underlines how the holiday weekends enjoy more contained effects of emission reductions compared to generic weekends of the year. Third, as to be expected, winds blowing from the West (Q${}_{\mathrm{SW}}$ and Q${}_{\mathrm{NW}}$) greatly reduce the amount of pollutants all over the city with peaks over 40% to 50%.

The short-term impacts adjusted for common anthropic and weather factors are summarized in Table 6, which shows the estimated permanent and transitory effects for each station in Milan expressed in logarithmic scale, hence interpretable as relative variations in concentrations levels. None of these two coefficients identifies an improvement of the considered pollutant concentrations. Moreover, the permanent effect ($\delta_{1}$) is always positive and in some cases moderately statistically significant. This means that, compared to the generally decreasing areal trend, Milan air quality went worst. It is worth observing that the most significant results are obtained at the Senato station, which is located in the already existing Area C, hence already subject to some car traffic restrictions.

Such a result could be justified by the presence of multiple causes. As a first justification, we are approaching the initial phase of a progressive policy and the time elapsed since its outset may be too short to assess any significant impacts on pollutant levels. This fact is consistent with the forecasts expected by the municipality of Milan about the reductions in nitrogen oxide levels; in fact, the first significant reductions should be observed starting from 2022 [11]. Furthermore, since we are dealing with limitations to human behavior and social perception of new norms, it is not always clear how agents adapt to changes. The deterioration in the air quality of the centre could be linked to new traffic congestions in that area or to a panic shock of drivers, who need time to understand the functioning of the restrictions and adapt their behavior, exactly as in situations mismanagement of individual and organizational changes [41,42]. Eventually, the recent climate changes and the extreme weather conditions that affected the Po valley, such as temperatures higher than the seasonal average and extreme atmospheric events, could increase the noise present in the data and thus mask the real repercussions of the limitations.

## 4. Conclusions

This paper analyzed the early-stage effects on air quality of the new traffic policy in Milan, the so-called Area B. The concentrations of nitrogen oxides (NO${}_{x}$) and nitrogen dioxide (NO${}_{2}$), which are mainly primary pollutants, have been considered as proxies of pollution emissions.

The first hypothesis in the introduction inquires about the presence of a significant effect on the air quality of the city. As a first point, the preliminary results show that concentrations during spring and summer 2019 are lower than during the same seasons in the previous five years, hinting for a reduction effect due to the policy. On the other side, a similar reducing trend has been observed in various neighbouring cities around Milan, which belong to a homogeneous meteorological, social, and economical cluster within the Po valley. Their similar behavior is used here as an areal common trend capturing both weather and anthropogenic components. Our approach, which adjusts for local weather conditions and the areal common trend, does not provide a further reduction effect for any station comparing to this trend. Instead, in Senato station, which is inside the historical city centre and was already covered by Area C, the estimates provide a strong, but moderately statistically significant, increase for both considered pollutants. This is coherent with the fact that the restriction introduced is very limited as it concerns just some classes of old vehicles, which are a small percentage of the entire vehicle pool, both in terms of number of cars and emissions.

Since the first research hypothesis is confirmed just to a minor extent and with an opposite sign with respect to what was expected, the second research hypothesis, concerning the homogeneity of the possible effects, assumes now only a technical scope. It is confirmed just for what concerns the positive direction of the changes, but not for their significance. In fact, among all the estimated permanent effects, only Senato station is significant at 5%. Moreover, the estimated transitory effects are always not significant at any confidence level.

The above facts hint that, compared to the common trend of the considered area, Milan air quality is improving slowly, and, in this sense, the first phase of Area B seems to have a negative effect on air quality. Due to the limited scope of this first phase and its progressiveness, it is not unexpected to find a limited or a zero effect. Nonetheless, the negative effect needs some more explanations.

Although finding the ultimate motivation for this is not the aim of this paper, a discussion follows. Firstly, the statistically significant increase found is limited in space and is located inside the previously introduced restricted Area C. It may be possible that this further restriction increased congestion of public transport buses, which are often very old vehicles, or to the aforementioned adaptation shocks. This could explain only a part of the results. In fact, this first point is also related to the other sources of nitrogen oxides. According to INEMAR [25], road traffic is about 68% of the total emissions. Hence, a transition to house heating green techniques slower in Milan comparing the other considered cities could have an influence on this result. Moreover, also the other stations experienced a comparative deterioration of air quality and the second-worst station is Città Studi, which is an urban background station, hence with limited relation to local traffic congestion. Second, the increase due to road traffic may have temporal dynamics. Since the traffic restriction is limited to business hours, there may be an increase in congestion early in the morning and in the late evening, affecting the daily average.

In conclusion, although environmental protection policies are in general a fundamental step for sustainability improvement, in some cases, they may not be sufficient or their implementation may be misleading. In our case, we considered only the early-stage of a policy, which is progressive in time. Hence, the results of this paper may be regarded as physiological, provided that they characterize only the initial part of the policy implementation and are improved soon. It follows a recommendation to the municipal government to develop the policy more strongly.

Additional research could be developed in the future. In particular, the effect on traffic congestion inside Area C could be investigated further using historical data related to the vehicle movements crossing the access points. Moreover, the use of a multivariate approach, which includes other pollutants such as PM${}_{10}$ and PM${}_{2.5}$, and spatio-temporal modeling could highlight hidden effects, which are not visible considering the single stations. Eventually, the extension to hourly data could consider both the presence of intra-daily effects and explaining the spatial dynamics related to traffic.

## Figures and Tables

**Figure 1 ijerph-17-01088-f001:**
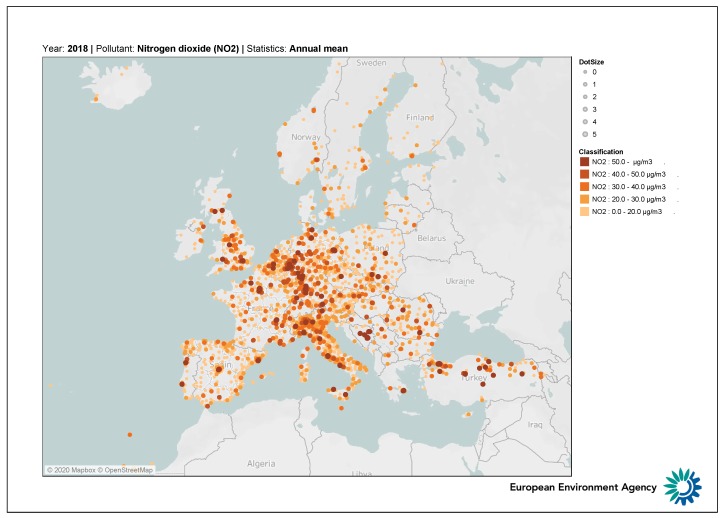
Annual average NO${}_{2}$ concentrations (μg/m${}^{3}$) in Europe during 2018. Levels are expressed in μg/m${}^{3}$. Source: European Environmental Agency

**Figure 2 ijerph-17-01088-f002:**
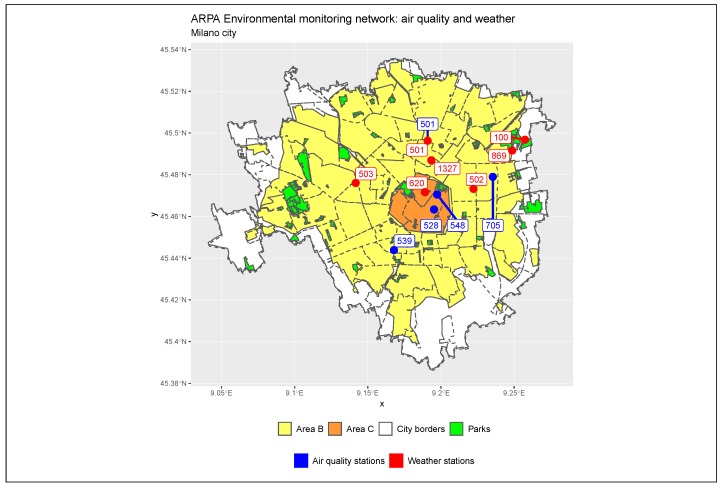
Monitoring system in Milan. Air quality stations (blue points): Marche (501), Verziere (528), Senato (548), Liguria (539) and Città Studi (705). Weather stations (red points): Lambrate (100), Zavattari (503), Brera (620), Feltre (869), Rosellini (1327), Juvara (502), and Marche (501).

**Figure 3 ijerph-17-01088-f003:**
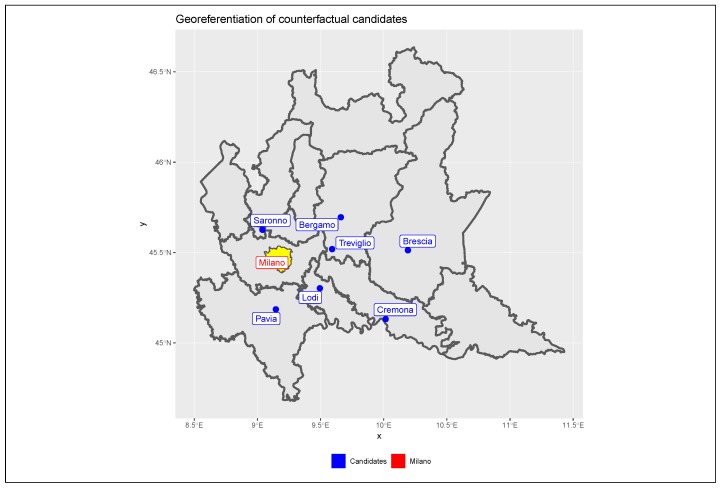
Georeferentiation of counter-factual candidates. Geographical positioning of the counter- factual candidates with respect to Milan.

**Figure 4 ijerph-17-01088-f004:**
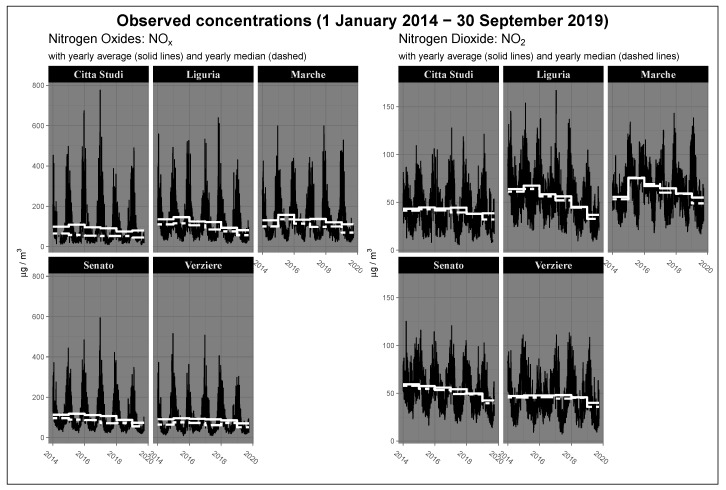
Pollutant levels in Milan (μg/m${}^{3}$). Observed concentrations levels of NO${}_{x}$ and NO${}_{2}$ between 2014 and 2019 with yearly average and median values. Values are expressed as μg/m${}^{3}$.

**Figure 5 ijerph-17-01088-f005:**
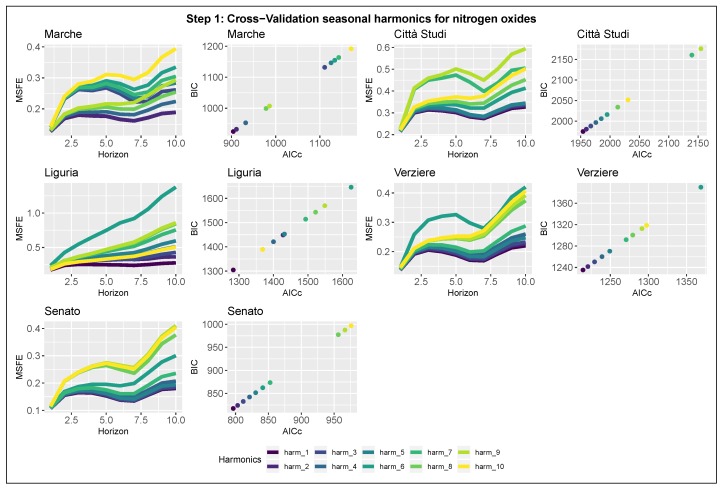
Model selection-Step 1-NO${}_{x}$. Seasonal component selection for the five nitrogen oxides stations in Milan. Left panel: 10-steps-ahead MSFE in log-scale as function of the number of harmonics. Right panel: AICc and BIC pairs for each model.

**Figure 6 ijerph-17-01088-f006:**
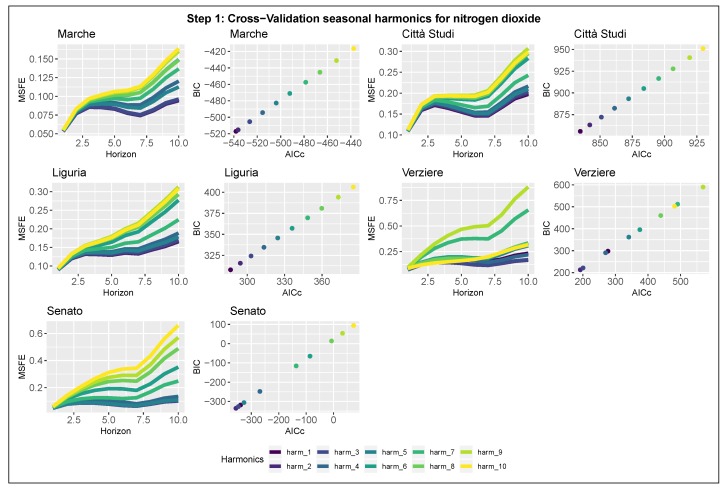
Model selection-Step 1-NO${}_{2}$. Seasonal component selection for the 5 nitrogen dioxide stations in Milan. Left panel: 10-steps-ahead MSFE in log-scale as function of the number of harmonics. Right panel: AICc and BIC pairs for each model.

**Figure 7 ijerph-17-01088-f007:**
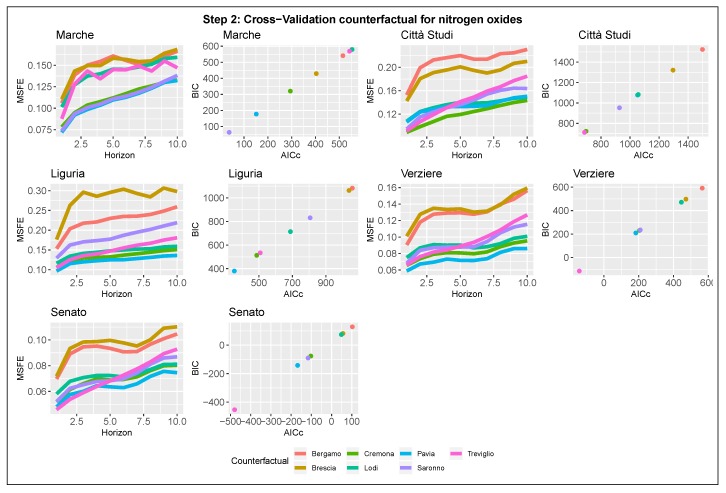
Model selection-Step 2-NO${}_{x}$. Counter-factual term selection for the five nitrogen oxides stations in Milan. Left panel: 10-steps-ahead MSFE in log-scale as function of the candidate. Right panel: AICc and BIC pairs for each model.

**Figure 8 ijerph-17-01088-f008:**
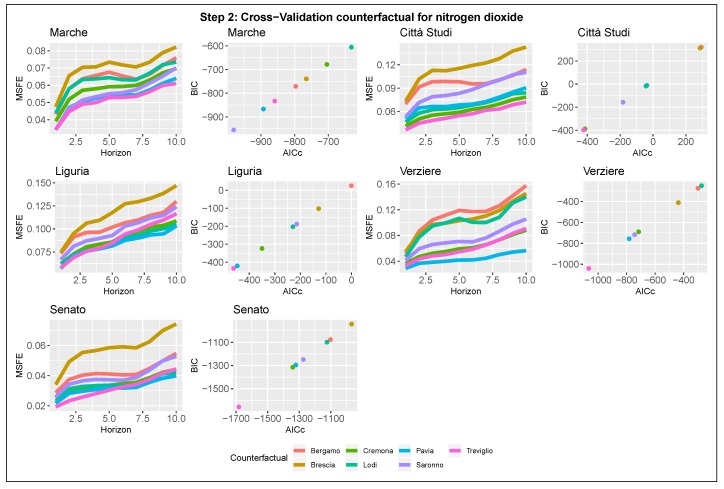
Model selection-Step 2-NO${}_{2}$. Counter-factual term selection for the five nitrogen dioxide stations in Milan. Left panel: 10-steps-ahead MSFE in log-scale as function of the candidate. Right panel: AICc and BIC pairs for each model.

**Table 1 ijerph-17-01088-t001:** Differences between the average concentration level of the sub-period 2014–2018 and the treatment period 2019. Differences are expressed in μg/m${}^{3}$.

Station Name	Nitrogen Oxides		Nitrogen Dioxide	
	$d_{\mathrm{AVG}}$	$d_{\mathrm{MED}}$	$d_{\mathrm{AVG}}$	$d_{\mathrm{MED}}$
*Milano city stations*				
Città Studi	0.94	−0.43	−2.63	−4.26
Liguria	−25.59	−26.91	−19.49	−20.71
Marche	−17.29	−22.00	−11.53	−13.37
Senato	−14.99	−13.84	−10.46	−9.73
Verziere	−2.66	−5.00	−4.35	−5.78
*Other urban centres in Lombardy*				
Bergamo	−11.56	−8.90	−5.11	−4.07
Brescia	−4.61	−6.39	−3.46	−4.12
Cremona	3.47	1.10	1.53	0.85
Lodi	−5.99	−7.62	-1.85	−2.57
Pavia	−14.75	−17.77	−7.75	−9.28
Saronno	−7.96	−7.84	−8.66	−8.87
Treviglio	0.06	−3.71	2.25	0.39

**Table 2 ijerph-17-01088-t002:** Model selection-Step 3-NO${}_{x}$: Best subset of covariates using backward-forward stepwise algorithms for NO${}_{x}$.

	Marche		Verziere		Senato		Liguria		Citta Studi	
	*AIC*	*BIC*	*AIC*	*BIC*	*AIC*	*BIC*	*AIC*	*BIC*	*AIC*	*BIC*
Holidays	✓	✓	✓	✓	✓	✓	✓	✓	✓	✓
Week-End	✓	✓	✓	✓	✓	✓	✓	✓	✓	✓
Saturday:Holidays	✓		✓		✓		✓		✓	
Sunday:Holidays	✓	✓	✓	✓	✓	✓	✓	✓	✓	✓
Wind speed Q${}_{\mathrm{NE}}$										
Wind speed Q${}_{\mathrm{SE}}$							✓			
Wind speed Q${}_{\mathrm{SW}}$	✓	✓	✓	✓	✓	✓	✓	✓	✓	✓
Wind speed Q${}_{\mathrm{NW}}$	✓	✓	✓				✓		✓	✓
Temperature										
Rainfall	✓		✓		✓				✓	
Global radiation			✓		✓					
Humidity										

*Note*: symbol ✓ indicates that the regressor is selected within the best subset of covariates.

**Table 3 ijerph-17-01088-t003:** Model selection-Step 3-NO${}_{2}$: Best subset of covariates using backward-forward stepwise algorithms for NO${}_{2}$.

	Marche		Verziere		Senato		Liguria		Citta Studi	
	*AIC*	*BIC*	*AIC*	*BIC*	*AIC*	*BIC*	*AIC*	*BIC*	*AIC*	*BIC*
Holidays	✓	✓	✓	✓	✓	✓	✓	✓	✓	✓
Week-End	✓	✓	✓	✓	✓	✓	✓	✓	✓	✓
Saturday:Holidays	✓		✓		✓		✓	✓	✓	
Sunday:Holidays	✓	✓	✓	✓	✓	✓	✓		✓	✓
Wind speed Q${}_{\mathrm{NE}}$										
Wind speed Q${}_{\mathrm{SE}}$										
Wind speed Q${}_{\mathrm{SW}}$	✓	✓	✓	✓	✓	✓	✓	✓	✓	✓
Wind speed Q${}_{\mathrm{NW}}$	✓								✓	
Temperature										
Rainfall			✓		✓					
Global radiation	✓		✓		✓				✓	
Humidity	✓								✓	

*Note*: symbol ✓ indicates that the regressor is selected within the best subset of covariates.

**Table 4 ijerph-17-01088-t004:** ML estimates of BSM parameters and variances for NO${}_{x}$.

	Parameter	*Marche*	*Citta Studi*	*Liguria*	*Verziere*	*Senato*
*log(Pavia)*	θ	0.51 ***	0.93 ***	0.73 ***	0.66 ***	0.57 ***
		(0.01)	(0.02)	(0.02)	0.02	(0.01)
*Holidays*	$\phi_{1}$	−0.06 ***	−0.02	−0.05 *	−0.10 ***	−0.09 ***
		(0.03)	(0.04)	(0.03)	(0.03)	(0.03)
*WeekEnd*	$\phi_{2}$	−0.11 ***	−0.09 ***	−0.09 ***	−0.17 ***	−0.15 ***
		(0.01)	(0.02)	(0.01)	(0.01)	(0.01)
*Saturday:Holidays*	$\phi_{3}$	0.10	0.17	0.04	0.10	0.14 ***
		(0.08)	(0.13)	(0.10)	(0.14)	(0.05)
*Sunday:Holidays*	$\phi_{4}$	0.09 **	0.09	0.12 *	0.08	0.10 ***
		(0.05)	(0.08)	(0.06)	(0.05)	(0.03)
*WindSpeed* Q${}_{\mathrm{SW}}$	$\phi_{5}$	−0.44 ***	−0.20 ***	−0.56 ***	−0.30 ***	−0.27 ***
		(0.01)	(0.02)	(0.02)	(0.02)	(0.01)
*WindSpeed* Q${}_{\mathrm{NW}}$	$\phi_{6}$	−0.31v***	−0.27 ***	−0.12 ***	−0.20 ***	−0.12 ***
		(0.01)	(0.02)	(0.01)	(0.01)	(0.01)
Level variance	$\sigma_{\eta }^{2}$	0.0047	0.0065	0.0037	0.0038	0.0027
Slope variance	$\sigma_{\zeta }^{2}$	0.0000	0.0000	0.0000	0.0000	0.0000
Seasonality variance	$\sigma_{\omega }^{2}$	0.0000	0.0000	0.0000	0.0000	0.0000
Error variance	$\sigma_{\varepsilon }^{2}$	0.0298	0.0745	0.0437	0.0370	0.0308

*Note 1*: values in parenthesis are standard errors. *Note 2*: * *p* < 0.10, ** *p* < 0.05, *** *p* < 0.01.

**Table 5 ijerph-17-01088-t005:** ML estimates of BSM parameters and variances for NO${}_{2}$.

	Parameter	*Marche*	*Citta Studi*	*Liguria*	*Verziere*	*Senato*
*log(Pavia)*	θ	0.36 ***	0.85 ***	0.69 ***	0.65 ***	0.55 ***
		(0.01)	(0.02)	(0.02)	(0.02)	(0.01)
*Holidays*	$\phi_{1}$	−0.06 ***	−0.08 ***	−0.08 ***	−0.10 ***	−0.08 **
		(0.02)	(0.03)	(0.02)	(0.02)	(0.03)
*Week-end*	$\phi_{2}$	−0.06 ***	−0.10 ***	−0.09 ***	−0.14 ***	−0.11 ***
		(0.01)	(0.01)	(0.01)	(0.01)	(0.01)
*Saturday:Holidays*	$\phi_{3}$	0.06	0.18 ***	0.06	0.14 **	−0.09 ***
		(0.05)	(0.08)	0.07	(0.06)	(0.02)
*Sunday:Holidays*	$\phi_{4}$	0.08 ***	0.11 ***	0.11 ***	0.06	0.10 ***
		(0.03)	(0.05)	(0.04)	(0.03)	(0.03)
*WindSpeed* Q${}_{\mathrm{SW}}$	$\phi_{5}$	−0.35 ***	−0.13 ***	−0.42 ***	−0.23 ***	−0.18 ***
		(0.01)	(0.01)	(0.01)	(0.01)	(0.01)
*WindSpeed* Q${}_{\mathrm{NW}}$	$\phi_{6}$	−0.17 ***	−0.16 ***	−0.08 ***	−0.13 ***	−0.08 ***
		(0.01)	(0.01)	(0.01)	(0.01)	(0.01)
Level variance	$\sigma_{\eta }^{2}$	0.0051	0.0065	0.0046	0.0044	0.0024
Slope variance	$\sigma_{\zeta }^{2}$	0.0000	0.0000	0.0000	0.0000	0.0000
Seasonality variance	$\sigma_{\omega }^{2}$	0.0000	0.0000	0.0000	0.0000	0.0000
Errors variance	$\sigma_{\varepsilon }^{2}$	0.0093	0.0282	0.0182	0.0154	0.0114

*Note 1*: values in parenthesis are standard errors. *Note 2*: ** *p* < 0.05, *** *p* < 0.01.

**Table 6 ijerph-17-01088-t006:** Estimated permanent and transitory effects in log scale on NO${}_{x}$ and NO${}_{2}$ for each station.

Stations	Effect	Nitrogen Oxides			Nitrogen Dioxide		
		Estimate	S.E.	*t*-Statistic	Estimate	S.E.	*t*-Statistic
Senato	Perm. eff. $\delta_{1}$	0.38	0.19	2.03 **	0.29	0.12	2.40 **
	Trans. eff. $\delta_{0}$	−0.12	0.16	−0.76	−0.14	0.11	−1.25
Verziere	Perm. eff. $\delta_{1}$	0.26	0.21	1.27	0.22	0.15	1.50
	Trans. eff. $\delta_{0}$	−0.01	0.18	−0.05	−0.06	0.14	−0.40
Liguria	Perm. eff. $\delta_{1}$	0.12	0.22	0.54	0.20	0.16	1.25
	Trans. eff. $\delta_{0}$	−0.02	0.19	−0.10	−0.10	0.15	−0.67
Marche	Perm. eff. $\delta_{1}$	0.15	0.19	0.80	0.23	0.13	1.82 *
	Trans. eff. $\delta_{0}$	−0.11	0.17	−0.63	−0.19	0.13	1.56
Citta Studi	Perm. eff. $\delta_{1}$	0.35	0.29	1.19	0.25	0.19	1.30
	Trans. eff. $\delta_{0}$	0.08	0.25	0.30	0.02	0.18	0.10

*Note 1*: * *p* < 0.10, ** *p* < 0.05.

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
