# Peer review of "Statistical Modeling of the Early-Stage Impact of a New Traffic Policy in Milan, Italy"

_ijerph, 2020, doi:10.3390/ijerph17031088_

Round 1

Reviewer 1 Report

The discussion section can be expanded

Author Response

Dear reviewer,

Thank you for your report. In the following lines, we will report the answers to your suggestions point-by-point. Please, note the attached file in which You will find the complete rebuttal letter for the editors and all the reviewers.

In the attached rebuttal, we answer point-to-point to all comments. For each comment of the reviewer we reported the original text in black, our response to the comment in red and the changes to manuscript in blue. Note that in the revision of the manuscript, all changes are in red and easily identifiable.
In summary, the new version of the manuscript includes the following revisions:
   • We updated substantially the abstract;
   • We changed the title to “Statistical modelling of the early-stage impact of a new traffic policy in Milan, Italy”;
   • We included a new short subsection about the software (Sec. 2.3.4 – Software, rows 343-347).

Sincerely yours,
Paolo Maranzano, Alessandro Fassò, Matteo Pelagatti and Manfred Mudelsee

-----------------------------------------------------------------------------

REVIEWER: The discussion section can be expanded
RESPONSE: Good suggestion! Accordingly with the requests of the other reviewers, we integrated the final discussion specifying better some aspects. First, we underlined the role of the counterfactual term in our models, specifying the criteria followed to identify it and to select it (territorial and socio-economic homogeneity, etc.). Second, we expanded the debate on how much the nitrogen oxides are relevant for the air quality in the Po Valley area and what the causes of their excesses could be. Additionally, we considered the consequences our results for policy makers and proposed some recommendations. Finally, we specified possible further research developments, which include the use of PM and other pollutants and the implementation of spatio-temporal modelling.
Note that further discussion is included in the ‘Results’ section (Sec. 3.3) as it is closely linked to the empirical results. That part has been integrated comparing our estimates with the expected forecast provided by the Municipality of Milan.
CHANGES TO MANUSCRIPT: The discussion relative to the empirical results has been integrated including the expected forecast provided by the Municipality of Milan (Sec. 3.3 – BSM & policy intervention, rows 454-456). The counterfactual role has been better specified (Sec. 4 – Discussion, rows 473-477). The possible causes of air quality degradation in Milan area have been expanded (Sec. 4 – Discussion, rows 497-499). Policy maker recommendations have been added to
the discussion section (Sec. 4 – Discussion, rows 505-510). Future works have been declared and justified (Sec. 4 – Discussion, rows 513-515).

Reviewer 2 Report

Reviewer:

The manuscript make a statistical analysis to evaluate impact of new traffic policy in Milan. It is a relevant research with some important results. The main concern is that hourly data was not used and for that reason the dynamics related to traffic are not considered. I do however believe there is still room for improvement and would like to provide the following comments:

My comments:

Abstract: The abstract is clear. However, you should mention the Euro class of the vehicles that are affected. Introduction: The references are updated. However, some additional references can be included to support the work. Line 26: Could be important to make a reference do Integrated assessment modelling (IAM) tools . Lines 50-1: mention “a small percentage of the entire vehicle pool”, it is not possible to obtain an estimation of the number? Figure 1: It is difficult to read and is just a print screen of the website. Why the authors not considered using also PM10/PM2.5? They are also related to road traffic. For me is difficult to understand why other urban centers in Lombardy (e.g. Brescia) are considered if traffic is restricted in Milan. The justifications given for the obtained results should better explained as well as the study limitations.

Author Response

Dear reviewer,

Thank you for your report. In the following lines, we will report the answers to your suggestions point-by-point. Please, note the attached file in which You will find the complete rebuttal letter for the editors and all the reviewers.

In the attached rebuttal, we answer point-to-point to all comments. For each comment of the reviewer we reported the original text in black, our response to the comment in red and the changes to manuscript in blue. Note that in the revision of the manuscript, all changes are in red and easily identifiable.
In summary, the new version of the manuscript includes the following revisions:
   • We updated substantially the abstract;
   • We changed the title to “Statistical modelling of the early-stage impact of a new traffic policy in Milan, Italy”;
   • We included a new short subsection about the software (Sec. 2.3.4 – Software, rows 343-347).

Sincerely yours,
Paolo Maranzano, Alessandro Fassò, Matteo Pelagatti and Manfred Mudelsee

-----------------------------------------------------------------------------

REVIEWER: The manuscript make a statistical analysis to evaluate impact of new traffic policy in Milan. It is a relevant research with some important results. The main concern is that hourly data was not used and for that reason the dynamics related to traffic are not considered.
RESPONSE: We thank you for the comment and we are in complete agreement on the possibility of using hourly instead of daily data. This paper is a preliminary study on the early-stage impact of a progressive policy. Our future goal is to deepen the topic in further research using various techniques, among which the approach with hourly data and spatio-temporal modelling are certainly the main candidates.
CHANGES TO MANUSCRIPT: We discussed our future perspectives, including the possibility to use spatio-temporal models, in the final section (Sec. 4 – Conclusion, rows 499-516).

REVIEWER: I do however believe there is still room for improvement and would like to provide the following comments: My comments:
Abstract: The abstract is clear. However, you should mention the Euro class of the vehicles that are affected.
RESPONSE: Thank you for this note. The original version did not include any details about the involved classes of vehicles due to a limitation on the number of words in the abstract (max 200 words). After a brief talk with the editor we were allowed to increase its length and it is now included.
CHANGES TO MANUSCRIPT: We mentioned in the abstract the same class of vehicles cited in the text (see Sec.1 - Introduction).

REVIEWER: Introduction: The references are updated. However, some additional references can be included to support the work. Line 26: Could be important to make a reference to Integrated assessment modelling (IAM) tools.
RESPONSE: Thank you for this suggestion; we investigated the literature about IA models and we integrated the text with some new references.
CHANGE TO MANUSCRIPT: We introduced the IA models citing some related articles (Sec.1 - Introduction, rows 30-36).

REVIEWER: Lines 50-1: mention “a small percentage of the entire vehicle pool”, it is not possible to obtain an estimation of the number?
CHANGE TO MANUSCRIPT: We inserted the estimation of the share of involved vehicles provided by the municipality of Milan. We also cited the regulatory reference that introduces and explains the structure of the policy (Sec. 1 – Introduction, rows 59-65).

REVIEWER: Figure 1: It is difficult to read and is just a print screen of the website.
CHANGE TO MANUSCRIPT: Figure 1 has been replaced with a new pdf-format figure (vector graphic) directly downloaded by the EEA website using their web dashboard (https://www.eea.europa.eu/data-and-maps/dashboards/air-quality-statistics).

REVIEWER: Why the authors not considered using also PM10/PM2.5? They are also related to road
traffic.
RESPONSE: Really interesting point. The main reason that led us to consider nitrogen oxides lies in some specific characteristics of air quality in the city of Milan. According to INEMAR - ARPA Lombardy emission inventory 2014, in Milan province, 68% of NOx and only 41% of PM10 are due to road traffic. Hence, in this first study of Area B, we decided to take into account nitrogen oxides
concentrations (NOx) and NO2 and postpone the analysis of PM10 and PM2.5 to further research.
CHANGE TO MANUSCRIPT: We motivated our decision to analyze NOx and NO2 in the introduction (Sec. 1 – Introduction, rows 90-96) and in the conclusive section (Sec. 4 – Conclusion, rows 496- 499). We also added a reference of the INEMAR emission inventory.

REVIEWER: For me is difficult to understand why other urban centers in Lombardy (e.g. Brescia) are considered if traffic is restricted in Milan.
RESPONSE: Thank you for addressing this important topic that lies at the basis of the paper. To identify correctly the traffic policy effect, we followed a treatment-control approach for time series which requires a counterfactual term. In our case, we proposed seven important urban centers located in the Lombardy Po Valley area, which show socio-demographic and economic
characteristics and weather conditions similar to Milan, but which are not directly affected by the considered policy. The second step of model selection procedure aims to select the best in terms of predictive power and estimation quality, able to capture weather conditions and socioeconomic features common to the area surrounding Milan in the Po valley.
CHANGE TO MANUSCRIPT: We discuss these facts both in the section Data & Method (Sec. 2 – Data, rows 188-198) and in the conclusive section (Sec. 4 – Conclusion, rows 473-478).

The justifications given for the obtained results should better explained as well as the study limitations.
RESPONSE: Completely agree with you, thank you. Following your suggestions and those of the other reviewers, the discussion of empirical results was integrated. The discussion about the empirical results has been integrated comparing our estimates with the expected forecast provided by the Municipality of Milan. We also expanded the debate on how much the nitrogen oxides are relevant for the air quality in the Po Valley area and what the causes of their excesses could be. The shift towards the concepts of ‘early-stage impact’ and ‘first phase of the policy’, as well as towards that of ‘progressive policy’, shows how this research can be developed in a broad and lasting way despite the initial limitations due to the regulation. Additionally, we considered the consequences our results for policy makers and proposed some recommendations.
CHANGES TO MANUSCRIPT: The discussion relative to the empirical results has been integrated including the expected forecast provided by the Municipality of Milan (Sec. 3.3 – BSM & policy intervention, rows 454-456). The sources of NOx in Milan area have been expanded (Sec. 4 – Discussion, rows 497-499). ‘Early-stage impact’, ‘first phase policy’ and ‘progressive policy’ topics are treated in the abstract, in the introduction (Sec.1 – Introduction, rows 49-63) and within the
conclusion (Sec.4 – Discussion, rows 488-491). Policy maker recommendations have been added to the discussion section (Sec. 4 – Discussion, rows 505-510).

Reviewer 3 Report

Regarding the manuscript entitled “Statistical modeling of the short-term impact for a new traffic policy in Milan, Italy”. The study is very interesting and the methodology used is scientifically sound. In my opinion the most interesting part of the study is the methodological approach that it is used and its presentation, because in my opinion the time period that has passed since the implementation of the traffic policy is too small to fully assess its effect. My only objection is that even though the authors explain in the text adequately enough why the new traffic policy has a negative impact on air quality, that is not clear in the abstract which might be very misleading to the reader and create a bad impression about policy efforts. It should be stated clearly in the text and in the abstract, that the time period that has passed since the implementation of the measure is very limited to draw safe conclusions for the long term results and more importantly that there is no information regarding the effect on other pollutants such as PMx which might be positive. I acknowledge that the effect of the measure might be negative even in long term for NOx, but for one thing this is not certain and second it might be positive for other pollutants. Both of these facts should be clearly mentioned in the manuscript and abstract, not to support the policy making, but not to mislead both the readers and the policy makers.

Author Response

Dear reviewer,

Thank you for your report. In the following lines, we will report the answers to your suggestions point-by-point. Please, note the attached file in which You will find the complete rebuttal letter for the editors and all the reviewers.

In the attached rebuttal, we answer point-to-point to all comments. For each comment of the reviewer we reported the original text in black, our response to the comment in red and the changes to manuscript in blue. Note that in the revision of the manuscript, all changes are in red and easily identifiable.
In summary, the new version of the manuscript includes the following revisions:
• We updated substantially the abstract;
• We changed the title to “Statistical modelling of the early-stage impact of a new traffic policy in Milan, Italy”;
• We included a new short subsection about the software (Sec. 2.3.4 – Software, rows 343-347).

Sincerely yours,
Paolo Maranzano, Alessandro Fassò, Matteo Pelagatti and Manfred Mudelsee

-----------------------------------------------------------------------------

REVIEWER: The study is very interesting and the methodology used is scientifically sound. In my opinion the most interesting part of the study is the methodological approach that it is used and its presentation, because in my opinion the time period that has passed since the implementation of the traffic policy is too small to fully assess its effect. My only objection is that even though the authors explain in the text adequately enough why the new traffic policy has a negative impact on air quality, that is not clear in the abstract which might be very misleading to the reader and create a bad impression about policy efforts.
RESPONSE: We thank you for this valuable comment, with which we are in perfect agreement. The original version was not exhaustive about the observed effects of the policy due to a limitation on the number of words in the abstract (max 200 words). After a brief talk with the editor we were allowed to increase its length.
CHANGE TO MANUSCRIPT: The abstract has been integrated including a short discussion about the structure of the policy, that means its progressiveness both in time and in the vehicle pool involved. We also discussed the fact that a short time window, such as the one at our disposal, may not be sufficient for social adaptation to the introduction of new rules and therefore would lead to unclear conclusions.

REVIEWER: It should be stated clearly in the text and in the abstract, that the time period that has passed since the implementation of the measure is very limited to draw safe conclusions for the long term results …
RESPONSE & CHANGE TO MANUSCRIPT: Again, thank you for reporting this fundamental aspect. The first version of the paper quickly discussed the problem of the limited time window available to us for policy evaluation, now we have extended the discussion. We shifted towards the concepts of ‘early-stage impact’, ‘first phase of the policy’, and ‘progressive policy’, showing how this research can be developed in a broad and lasting way despite the initial limitations due to the
regulation.
CHANGE TO MANUSCRIPT: An extended discussion about the possible causes of our results has been developed in section ‘Results’ (Sec. 3.3 – BSM and policy intervention, rows 452-464). We considered the length of the series and the time elapsed from the beginning of the first phase of the policy, as well as the human response to social adaptation. We clearly stated that the policy is now facing the preliminary phases and that our results are affected by this factor. The introduction of traffic restrictions are in any case to be considered positive for social well-being, but it takes time before having the desired effects. To further argue this position, we reported in the text the expected effects on pollutants for the next few years provided by the municipality of Milan (Sec. 1 – Introduction, rows 57-63). ‘Early-stage impact’, ‘first phase policy’ and ‘progressive policy’ topics are treated in the abstract, in the introduction (Sec.1 – Introduction, rows 49-63) and within the conclusion (Sec.4 – Discussion, rows 488-491).

REVIEWER: … and more importantly that there is no information regarding the effect on other pollutants such as PMx which might be positive. I acknowledge that the effect of the measure might be negative even in long term for NOx, but for one thing this is not certain and second it might be positive for other pollutants. Both of these facts should be clearly mentioned in the manuscript and abstract, not to support the policy making, but not to mislead both the readers
and the policy makers.

RESPONSE: Really interesting point. The main reason that led us to consider nitrogen oxides lies in some specific characteristics of air quality in the city of Milan. According to INEMAR - ARPA Lombardy emission inventory 2014, in Milan province, 68% of NOx and only 41% of PM10 are due to road traffic. Hence, in this first study of Area B, we decided to take into account nitrogen oxides
concentrations (NOx) and NO2 and postpone the analysis of PM10 and PM2.5 to further research. Additionally, we considered the consequences our results for policy makers and proposed some recommendations.
CHANGE TO MANUSCRIPT: We motivated our decision to analyze NOx and NO2 in the introduction (Sec. 1 – Introduction, rows 90-96) and in the conclusive section (Sec. 4 – Conclusion, rows 496-499). We also added a reference of the INEMAR emission inventory. Policy maker recommendations have been added to the discussion section (Sec. 4 – Discussion, rows 505-510)
